# Cerebral Overperfusion Despite Reduced Cortical Metabolism Is Associated with Postoperative Delirium in Cardiac Surgery Patients: A Prospective Observational Study

**DOI:** 10.3390/jcm13216605

**Published:** 2024-11-03

**Authors:** Marcus Thudium, Lara Braun, Annika Stroemer, Andreas Mayr, Jan Menzenbach, Thomas Saller, Martin Soehle, Evgeniya Kornilov, Tobias Hilbert

**Affiliations:** 1Department of Anaesthesiology, University Hospital Bonn, 53127 Bonn, Germany; mthu@uni-bonn.de (M.T.); lara.braun@ukbonn.de (L.B.); jan.menzenbach@ukbonn.de (J.M.); martin.soehle@ukbonn.de (M.S.); 2Department of Surgery, University Hospital Bonn, 53127 Bonn, Germany; 3Department of Medical Biometrics, Informatics and Epidemiology, University Hospital Bonn, 53127 Bonn, Germany; stroemer@imbie.uni-bonn.de (A.S.); amayr@uni-bonn.de (A.M.); 4Department of Anaesthesiology, Campus Grosshadern, University Hospital, LMU Munich, 81377 Munich, Germany; thomas.saller@med.uni-muenchen.de; 5Department of Neurobiology, Weizmann Institute of Science, Rehovot 7610001, Israel; doctor.kornilov@gmail.com

**Keywords:** postoperative delirium, cerebral blood flow, cerebral oximetry, transcranial Doppler sonography, bispectral index, cerebral autoregulation

## Abstract

**Background**: Decreased cerebral oximetry (rSO_2_) in cardiac surgery is associated with postoperative delirium (POD). However, interventions optimizing intraoperative rSO_2_ are inconclusive. **Methods**: In this prospective observational cohort study, the relationship between rSO_2_, middle cerebral artery blood flow velocity (MCAV), and processed EEG was assessed in cardiac surgery patients with and without POD. MCAV was continuously recorded by transcranial Doppler sonography (TCD), together with continuous rSO_2_ and bispectral index (BIS) monitoring. Cardiopulmonary bypass (CPB) flow rate was adjusted according to body surface area. The cohort was divided into the POD and control groups, according to the postoperative results of the confusion assessment method (CAM/CAM-ICU), the 4A’s test (4AT), and the Delirium Observation Scale (DOS). A mixed model analysis was performed for intraoperative raw data. The cerebral autoregulation index was calculated from TCD, rSO_2_, and arterial pressure values. Differences in impaired autoregulation were compared using the Mann–Whitney U test. **Results**: A total of 41 patients were included in this study. A total of 13 patients (36.11%) developed postoperative delirium. There were no significant differences in the baseline characteristics of patients with or without POD. Patients with POD had lower BIS values during CPB (adjusted mean difference −4.449 (95% CI [−7.978, −0.925])). RSO_2_ was not significantly reduced in POD, (adjusted mean difference: −5.320, 95% CI [−11.508, 0.874]). In contrast, MCAV was significantly increased in POD (10.655, 95% CI [0.491, 20.819]). The duration of cerebral autoregulation impairment did not differ significantly for TCD and cerebral oximetry-derived indices (*p* = 0.4528, *p* = 0.2715, respectively). **Conclusions**: Our results suggest that disturbed cerebral metabolism reflects a vulnerable brain which may be more susceptible to overperfusion during CPB, which can be seen in increased MCAV values. These phenomena occur irrespectively of cerebral autoregulation.

## 1. Introduction

Postoperative delirium (POD) and postoperative cognitive dysfunction are among the most common complications following cardiac surgery. Depending on patients’ comorbidities, a prevalence of up to 50% has been reported [1]. Since POD itself is known to prolongate intensive care unit (ICU) stay as well as hospital stay due to substantially increased postoperative morbidity and mortality [2], strategies to effectively prevent or treat postoperative cognitive decline are urgently needed. Moreover, its underlying pathogenetic mechanisms still remain poorly understood.

The association of POD with intraoperative EEG suppression and low bispectral index (BIS) values is well known and has been demonstrated in several studies [3,4,5]. However, whether there is a causal link between EEG-guided anesthesia and a reduced incidence of POD is still a subject of debate [5]. The recently proposed “vulnerable brain” hypothesis suggests that patients with pre-existing cognitive impairment are at risk and may be especially susceptible to anesthetics. This manifests itself in the loss of frontal alpha wave activity during anesthesia [6].

Arterial blood pressure above the upper cerebral autoregulation limits during cardiopulmonary bypass (CPB), determined by near-infrared spectroscopy (NIRS), has been proposed to contribute to the development of POD in cardiac surgery patients [7]. However, our previous results suggest that cerebral hyperperfusion is more dependent on CPB pump flow and therefore independent of cerebral autoregulation limits [8]. Integrating these findings points towards a certain pre-existing vulnerability of the brain, represented by atrophy and reduced cognitive reserve and the risk of cerebral overperfusion during CPB.

To shed further light on these hypotheses, we aimed to investigate the association of POD with processed EEG (BIS), NIRS, and transcranial doppler (TCD) sonography during CPB. In addition, we assessed NIRS- and TCD-derived autoregulation indices and their specific relation to POD. We hypothesized that there was an association between cerebral overperfusion as measured by TCD and POD irrespective of cerebral autoregulation.

## 2. Materials and Methods

### 2.1. Ethics

We performed this cross-sectional cohort study in accordance with the Declaration of Helsinki and after approval by the University of Bonn ethics committee (Chair: Prof. K. Racké, protocol numbers 300/16, 16 June 2016 and 355/17, 18 September 2017). Patients were enrolled after they provided written informed consent. The inclusion criteria were as follows: patient age >18 years and elective on-pump open heart surgery. The exclusion criteria were as follows: emergency procedure, aortic arch surgery with circulatory arrest, pregnancy, and the absence of a proper TCD window. Patients were recruited from October 2018 to April 2019.

### 2.2. Perioperative Management

All patients received anesthesia induction according to standard procedures including the use of arterial lines, intubation, central venous catheterization, and urinary catheters. Anesthesia was induced with sufentanil, etomidate, and rocuronium and was maintained with sevoflurane and the continuous infusion of sufentanil. The patients received noradrenaline as a vasopressor and dobutamine for inotropic support (the latter not during CPB). Median sternotomy was performed in apnea. After the systemic heparinization and cannulation of the ascending aorta and of the right atrium or the superior and inferior vena cava, respectively, CPB was established using a roller pump (Advanced Perfusion System 1, Terumo Corporation, Tokyo, Japan) and a membrane oxygenator (Quadrox-I Adult, Maquet Getinge Group, Rastatt, Germany). The heart–lung machine (HLM) system was primed with 1200 mL of crystalloid infusion solution and 10,000 IU of heparin. After full HLM support was installed with a pump flow rate of 2.5 L/m^2^ body surface area*min, pulmonary ventilation was stopped, and sevoflurane was continuously administered via the membrane oxygenator of the HLM system throughout the whole CPB period, with at least 0.7 of the minimum alveolar concentration (MAC) and not exceeding the maximum value of 1.5 MAC. Ventricular fibrillation was induced, the aorta was cross-clamped, and cardiac arrest was achieved by administering Calafiore warm blood or Bretschneider cardioplegia infusion. Mild hypothermia (32–34 °C) was induced, and acid–base management was performed according to the pH-stat regimen. Towards the end of the surgical procedure, patients were rewarmed and subsequently weaned from the CPB after the de-clamping of the aorta and sufficient reperfusion. After successful weaning from the CPB, heparin was antagonized with protamine in an 80–100% dose of the initially administered heparin. Patients were postoperatively transferred to the intensive care unit where they were extubated when stable circulation hemodynamic and pulmonary conditions were present.

### 2.3. Neuromonitoring and Data Acquisition

Continuous cerebral NIRS monitoring (rSO_2_) was established by attaching an optode above the left frontal lobe and using an INVOS 5100 C monitor (Covidien Medtronic, Dublin, Ireland). All patients received the monitoring of the bispectral index (BIS, BIS vista, Medtronic Minimally Invasive Therapies, Minneapolis, MN, USA). TCD was performed using a Delica 9UA Doppler ultrasound system (Shenzen Medical, Shenzen, China). The right transtemporal Doppler window was used to reach the middle cerebral artery at a small angle: the ultrasound transducer was placed on the temporal fossa slightly above the zygomatic arch. A pulsatile Doppler signal was localized in the depth of 3.5–4.5 cm in the M1 segment of the vessel. The depth of the signal was adjusted according to the strength of the middle cerebral artery flow velocity (MCAV) signal. The ultrasound probe was fixed in position using an adjustable headframe. Adjustments were made to the ultrasound transducer if signal quality was reduced during the procedure. Monitoring was performed following the induction of anesthesia and throughout the whole procedure until the end of surgical measures when patients were transferred from the operation room to the ICU. Only cerebral data obtained during the CPB period were used for statistical analysis in this study. MCAV values were recorded as mean flow velocity values with a sampling rate of 1 Hz.

### 2.4. Cerebral Autoregulation Assessment

For cerebral autoregulation assessment, a modified version of the Mx or COx algorithm was used [9,10]. Since low-resolution hemodynamic data were recorded, a modification of the low-resolution autoregulation index (LAx) using middle cerebral artery blood flow velocity and cerebral oxygen saturation was used for the calculation of Mx and COx, respectively [11].

### 2.5. Postoperative Delirium Screening

Delirium screening was performed as part of the PRe-Operative Prediction of postoperative DElirium by appropriate SCreening (PROPDESC) study. A positive POD diagnosis was considered if any of the applied assessment methods, specified below, detected POD at least once during the 5-day postoperative visit period. Delirium screenings were performed every morning by trained study personnel on each of the first five days after surgery or the first five days after the end of sedation. Sedated patients with a Richmond Agitation-Sedation Scale (RASS) score ≤3 were considered unable to be assessed, and POD testing was postponed to the following day. The Confusion Assessment Method for ICU (CAM-ICU) was used for intensive care patients [12]. The Confusion Assessment Method (CAM) and 4A’s test (4AT) incorporating Alertness, Abbreviated Mental Test-4, Attention (Month Backwards test), and acute change and fluctuating course were conducted on patients in standard wards [13]. In order to not miss the presence of delirium due to spot examinations conducted by the study staff, nurses in charge were interviewed at each visit using the Delirium Observation Screening Scale (DOS). POD diagnosis was rated positive from a 4AT score of 4 points or a DOS score of 3 and above [14]. Structured data and test results were entered pseudonymized into an electronic database.

### 2.6. Sample Size Estimation

Based on previous experiences from a historical cohort [8] and therefore assuming an MCAV of 90% of baseline values for the reference group and 110% in the POD group with a standard deviation of 18% (Cohen’s d of 1.11) with an assumed ratio reference for POD of 2:1, a sample size of 32 would be necessary to achieve a power of >80% and a level of significance of 5%. Further assuming a large dropout rate of 25% due to missing data, inability to obtain a TCD signal, or missing POD assessment, we aimed at a sample size of >40 patients.

### 2.7. Data Analysis

The exploratory analysis followed the STROBE guidelines for observational studies [15]. Since, according to the Shapiro–Wilk test, not all variables were normally distributed, data are presented as median values with interquartile ranges, and intergroup differences were evaluated using non-parametric testing. For descriptive statistical analyses of patient demographics, the Mann–Whitney U test was applied for continuous parameters, and Fisher’s Exact test was used for categorical variables. The alpha level was set to 0.05.

To analyze the relationship between rSO_2_, BIS, MAP, pump flow, and MCAV and the occurrence of POD while accounting for the longitudinal structure of the data, linear mixed effect models were used for each outcome separately. Each model incorporated the grouping variable (POD/control) and time as fixed effects, with subject-specific random effects to account for repeated measurements. Effect estimates with corresponding 95% confidence intervals are reported as adjusted group differences. For the calculation of the cerebral autoregulation index between rSO_2_ and MAP and MCAV and MAP, the Pearson correlation coefficient was calculated over 5 min intervals, which were sequentially shifted by one time point and then normalized. The Mann–Whitney U test was used to compare the minutes of correlation above the cut-off value of 0.3 between POD and the control group [9].

All analyses were conducted using R version 4.0.4. The datasets generated and analyzed during this study are available from the corresponding author on request.

## 3. Results

A total of 41 patients were initially included in this study. Five patients had to be excluded from further analysis since they had to be sedated or reintubated during the five postoperative days of POD testing, resulting in incomplete POD screening data. A total of 36 patients remained for the final analysis. Figure 1 shows a flowchart of patient inclusion, and Table 1 gives an overview on perioperative patient characteristics.

A total of 13 patients (36.11%) developed postoperative delirium. No significant difference between the baseline characteristics of patients developing POD compared to patients without POD could be found, as shown in Table 1. Effect estimates resulting from the mixed models for the group variable are shown in Table 2. Patients with delirium had significantly lower BIS values during CPB compared to patients without delirium, with an adjusted mean difference of −4.449 (95% CI [−7.978, −0.925]). Similarly, rSO_2_ was reduced in delirium patients, although the differences did not reach statistical significance (adjusted mean difference: −5.320, 95% CI [−11.508, 0.874]). In contrast, MCAV was significantly increased in delirium patients (10.655, 95% CI [0.491, 20.819]).

The Mann–Whitney U test showed no differences in minutes above the cerebral autoregulation threshold of 0.3 between the POD (107 min) and the non-POD group (99.67 min, *p* = 0.2715) for NIRS-derived autoregulation indices. TCD-derived autoregulation calculations showed similar results with 87.11 min of impaired autoregulation in patients without POD and 81.33 min with POD (*p* = 0.4528). Additionally, there was no difference between the no-POD and POD subcohorts in global hemodynamic parameters. Figure 2 shows global hemodynamic and cerebral parameters in the POD and non-POD groups. The results of the calculation of minutes above the autoregulation threshold for the two patient groups are presented in Figure 3.

## 4. Discussion

Postoperative delirium still remains a significant complication, and especially in cardiac surgery patients, the prevalence has been shown to be increased compared to other surgical disciplines. However, the causes of POD are still elusive, as are options to treat or prevent it.

The results of our present work suggest a mismatch between blood flow to the brain and its metabolic demands causing the development of POD. On the one hand, there are patients at risk presenting with a reduced cerebral metabolism, which is clinically reflected by reduced NIRS and BIS values. On the other hand, there is a non-adapted cerebral blood flow (CBF) during CPB, possibly exceeding the individuals’ requirements.

A given predisposition to develop POD has long been described [10]. It has also been shown that individuals with already impaired cognitive function, being the clinical correlate of decreased cerebral metabolism, possess a higher risk for POD [10]. Recently, research and clinical approaches to prevent POD development have focused on avoiding excessively deep anesthesia, especially in patients receiving only small doses of narcotics while reaching deep anesthesia and pronounced arterial hypotension [16]. A loss in alpha activity in the EEG frequency bands, together with burst suppression patterns, has also been shown to represent the brain’s vulnerability to develop delirium and requiring (or tolerating) only minimal doses of anesthetics [6]. The ability of the BIS to reflect cerebral metabolism has been suggested long ago [17]. While not focusing solely on EEG, our results generally appear to confirm these findings. Low BIS values in our POD cohort, together with a decreased rSO_2_, may represent a reduced cerebral metabolism. Such an association between NIRS and BIS in cardiac surgery patients has already been described before by Sugiura et al., supporting the assumption that both parameters are suitable to describe certain aspects of cerebral metabolism [18]. Furthermore, we previously described a relation between patient age and a reduced baseline MCAV assessed before CPB [8]. This may likewise reflect a reduced blood flow demand, caused by decreased brain metabolism and conversely increasing the risk of harmful cerebral overperfusion. Therefore, in physiological conditions, a decrease in BIS and rSO_2_ should be accompanied by an MCAV reduction as it has been described with the use of propofol [17]. Since this is not the case in our patients, we suggest that cerebral regulatory mechanisms are exceeded by the use of CPB. Surprisingly, patients developing POD showed no differences in either NIRS- and TCD-derived cerebral autoregulation indices during CPB, compared to the no-POD cohort. This is in contrast to the previous findings of Hori et al. who demonstrated that mean arterial blood pressure above the upper limit of autoregulation is associated with delirium [7]. Caldas et al. also reported an association between impaired cerebral autoregulation and POD. Although we could not reproduce these findings in our cohort, our results suggest a similar pathomechanistic approach for the development of POD following CPB, which is excess cerebral perfusion. Cerebral edema and brain swelling after CPB have already been shown, supporting this hypothesis [19,20]. However, MAP is not the only driving force for cerebral blood flow. Cardiac output has been shown to contribute to cerebral blood flow, independent from mean arterial blood pressure [21,22]. Therefore, even in intact cerebral autoregulation, cerebral overperfusion may occur as a result of inadequately increased CPB pump flow. Surprising on first view, in our patients, no intergroup differences could be observed in global hemodynamic parameters such as MAP or CPB pump flow rate. However, this is plausible when considering the individually different blood flow demands depending on cerebral metabolism [23].

Of course, our study has limitations. First, this is an observational study and should be interpreted cautiously in this context. Second, both NIRS and TCD are surrogate parameters with respective limitations. In NIRS, this is often the effect of extracranial contamination and in TCD the effect of possible changes in vessel diameter [24,25]. Indeed, the relationship between CBF and TCD-derived MCAV remains one important potential methodical limitation of our study. Since the first description of the use of TCD to assess CBF by Aaslid et al., numerous attempts to validate this method for various clinical situations (including CPB) have been made [26]. During a mild hypothermic CPB and pH-stat regimen, it can be assumed that at least changes in flow velocity in the MCA reflect changes in CBF with sufficient accuracy as long as arterial CO_2_ partial pressure and thus the diameter of the basal cerebral arteries remain constant [27,28]. Therefore, assessing cerebral perfusion as well as autoregulation indices during CPB using TCD appears to be valid. Another limitation of this study is the lack of baseline TCD data in awake patients, which makes it challenging to actually interpret changes in CBF. Last, hemodynamic data were only available with relatively low temporal resolution which may have influenced Mx and ORx calculations. While datasets with similar resolution were validated, this may still represent a source of error, and the results should be interpreted with caution.

In conclusion, we suggest a multi-dimensional approach, integrating both cerebral autoregulation assessment and TCD-derived cerebral blood flow velocity. Impaired autoregulation may be tackled by optimizing the mean arterial blood pressure, while excessive and harmful peaks of CBFV may be approached by incorporating an individually adjusted CPB pump flow rate. Whether such a strategy may help in preventing the development of POD remains to be investigated in further studies.

## Figures and Tables

**Figure 1 jcm-13-06605-f001:**
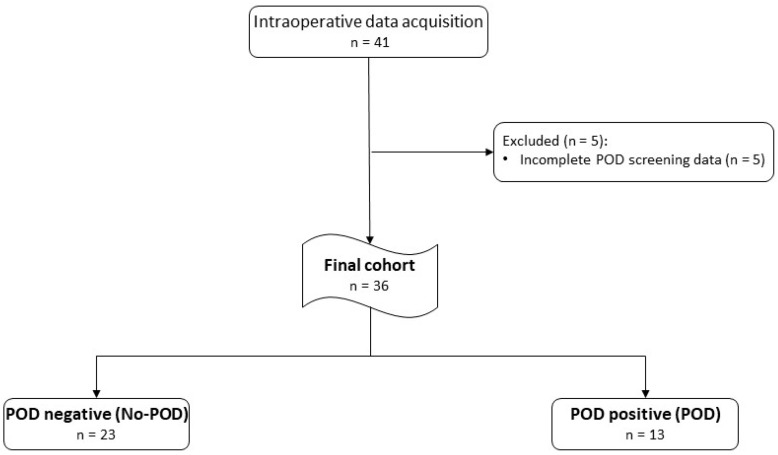
Study design and patient flowchart.

**Figure 2 jcm-13-06605-f002:**
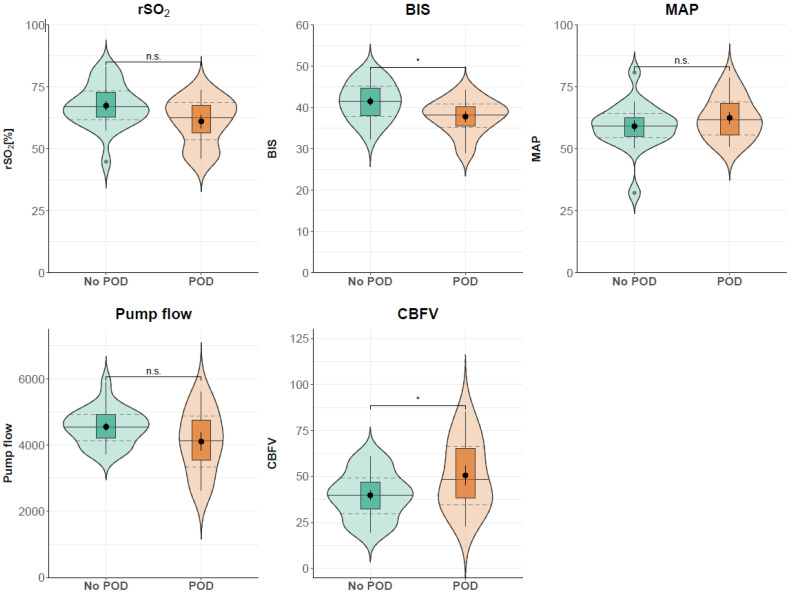
General hemodynamic and cerebral parameters in patients with and without delirium. Pump flow: blood flow of cardiopulmonary bypass; BIS: bispectral index; CBFV: (middle) cerebral artery mean blood flow velocity; MAP: mean arterial pressure; POD: postoperative delirium; rSO_2_: regional cerebral oxygen saturation. Median (solid lines) and interquartile range (dashed lines); * *p* < 0.05, n.s. = not significant (according to results from linear mixed effect models).

**Figure 3 jcm-13-06605-f003:**
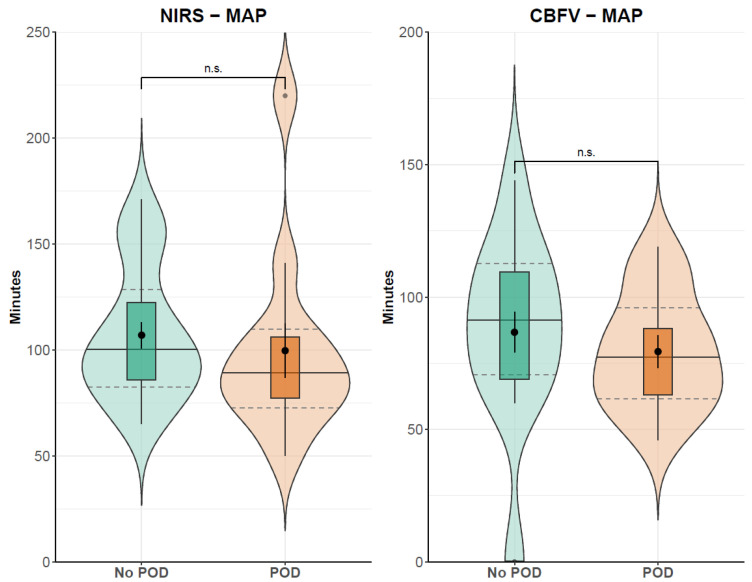
Absolute time of cerebral autoregulation indices above 0.3, indicating impaired cerebral autoregulation. Indices are derived from moving linear correlations between MAP and cerebral parameters. CBFV: (middle) cerebral artery mean blood flow velocity; MAP: mean arterial pressure; POD: postoperative delirium; rSO_2_: regional cerebral oxygen saturation. Median (solid lines) and interquartile range (dashed lines); n.s. = not significant (according to results from Mann–Whitney U test).

**Table 1 jcm-13-06605-t001:** Perioperative patient characteristics stratified by POD outcome.

Parameter	No POD	POD	*p* Value
Incidence (n)	23 (64%)	13 (36%)	
Age (years)	68 (13)	74 (12)	0.22
Male gender (n)	19 (83%)	9 (64%)	0.42 ^a^
Weight (kg)	89 (31)	77 (28)	0.14
Height (cm)	175 (13)	168 (21)	0.32
Body surface area (m^2^)	2.04 (0.43)	1.90 (0.41)	0.24
Body mass index (kg/m^2^)	28.2 (6.1)	27.8 (6.8)	0.30
Duration of bypass period (min)	135 (68)	135 (62)	0.40
Coronary artery bypass graft (n)	14 (61%)	6 (46%)	0.49 ^a^
Aortic valve repair (n)	4 (17%)	2 (15%)	0.99 ^a^
Mitral/Tricuspid valve repair (n)	1 (5%)	1 (8%)	0.99 ^a^
Combination surgery (n)	4 (17%)	4 (31%)	0.42 ^a^

Data are presented as median values with interquartile range. Mann–Whitney test and Fisher‘s Exact test (^a^) were used for analysis.

**Table 2 jcm-13-06605-t002:** Adjusted group differences with the corresponding 95% confidence intervals calculated via estimated effect of group variable (POD = 1/control group = 0) for different mixed models.

Parameter	Group Difference (Effect Estimate)	95% Confidence Interval
rSO_2_ [%]	−5.320	−11.508–0.874
BIS	−4.449	−7.978–−0.925
MAP [mmHg]	1.479	−5.389–8.263
PF [ml/min]	−279.470	−915.704–361.209
MCAV [cm/s]	10.655	0.491–20.819

rSO_2_: regional cerebral oxygen saturation; BIS: bispectral index; MAP: mean arterial pressure; PF: pump flow, blood flow of cardiopulmonary bypass; MCAV: (middle) cerebral artery mean blood flow velocity.

## Data Availability

The datasets generated and analyzed during this study are available from the corresponding author on request.

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
