# Peer review of "Cerebral Overperfusion Despite Reduced Cortical Metabolism Is Associated with Postoperative Delirium in Cardiac Surgery Patients: A Prospective Observational Study"

_jcm, 2024, doi:10.3390/jcm13216605_

Round 1
Reviewer 1 Report
Comments and Suggestions for Authors
Dear Authors
I read with great interest your paper "Cerebral overperfusion despite reduced cortical metabolism is
associated with postoperative delirium in cardiac surgery patients: A prospective observational study". Here, I have reported my suggestions to improve the quality of your work.
Introduction
This section is clear and well-written. The aim of the study is clearly stated.
Methods:
With which method did you administer sevoflurane during the on-pump phase?
Statistical consideration
1) How did you evaluate the normal distribution of continuous variables?
2) please add a cut-off p-value to consider statistically significant.
Result:
- 13 represents the incidence, not the prevalence, for POD. Furthermore, remove prevalence from Table 1.
Discussion
The authors addressed all findings derived from their analysis. Well done!
Reviewer 2 Report
Comments and Suggestions for Authors
The authors examined the relationship between cerebral oximetry (rSO2), middle cerebral artery blood flow velocity (MCAV), and EEG in cardiac surgery patients, focusing on postoperative delirium (POD). Among 41 patients, 13 (36.1%) developed POD. Patients with POD had lower bispectral index (BIS) values during cardiopulmonary bypass (CPB) and higher MCAV, while rSO2 was not significantly reduced. Cerebral autoregulation impairment duration did not differ significantly. The findings suggest POD may be linked to increased MCAV and disturbed cerebral metabolism during CPB.
General comments
This is a manuscript addressing an important topic “Cerebral overperfusion despite reduced cortical metabolism is associated with postoperative delirium in cardiac surgery patients: A prospective observational study”. However, the discussion and conclusions drawn are only partly supported by the results. Some concerns need to be addressed.
Specific comments
Major Comments
1) The timing of the assessment of cerebral parameters needs to be specified. The manuscript only mentions "until the end of surgery" (line 111).
2) The middle cerebral artery flow velocity (MCAV) signal may not accurately reflect blood flow because it depends on vascular resistance. More discussion is required on this point.
3) Lines 226-235: Related to point 1 above, in order to discuss excess blood flow in certain patients, baseline values are necessary to observe changes in cerebral parameters. Timeline data is also needed to support the conclusion that "Impaired autoregulation may be addressed by optimizing mean arterial pressure, while excessive and harmful peaks in CBFV may be managed through an individually adjusted CPB pump flow rate."
Minor Comments
1) Line 88: Replace [*] with [/].
2) Line 90: The aorta should be clamped before cardioplegia infusion.
3) Line 137: An explanation of the baseline values and the basic data used for the assumption is needed. If no basic data are available, please mention this.
4) Include the units for parameters in Table 2.
Reviewer 3 Report
Comments and Suggestions for Authors
Article
Cerebral overperfusion despite reduced cortical metabolism is associated with postoperative delirium in cardiac surgery patients: A prospective observational study
Authors present a prospective observational cohort study, where haemodynamic parameters were correlated with the possible development of postoperative delirium (POD) following cardiopulmonary bypass (CPB) surgery.
The main findings of this study include that patients with POD display:
· Reduced bispectral index (BIS)
· Unchanged rSO2 (left frontal lobe)
· Significantly increased middle cerebral artery blood flow velocity (doppler ultrasound)
From which authors conclude that overperfusion (increased MCAV) during CPB may be associated with development of POD
The main question from this reviewer is that blood flow velocity is not the same as volumetric blood flow. i.e. doppler ultrasound may show increased velocity while blood flow is actually decreasing (underperfusion). Did authors also assess cerebral artery diameter at the same time as velocity assessment? This could be used to give more accurate assessment of blood flow than simple velocity.
Questions/Suggestions
Line 42: "postoperative cognitive dysfunction (POCD)"
Comment: no need to define acronym that is never used again.
Line 44+: " reported.[1] "
Comment: citations should be listed before full stop. i.e. " reported [1]."
Line 45: suggest change "intensive care unit (ICU) " to
"intensive care unit (ICU) stay"
Line 46: "increased postoperative morbidity and mortality,"
Citation(s) needed.
Line 47: suggest change "urgently requested" to
"urgently needed"
Line 50: suggest change "is well known by now and has been" to
"is well known and has been"
Line 57: "However, our previous results suggest cerebral hyperperfusion depending on CPB pump flow and independent of cerebral autoregulation limits."
Comment: not complete sentence. Needs fixing for clarity.
Line 144: "STROBE guidelines for observational studies."
Citation(s) needed.
Line 195/22: Figures 2, 3:
please include in figure legend what lines indicate interquartile ranges, what indicates medians, what n.s. and * indicate.
Round 2
Reviewer 2 Report
Comments and Suggestions for Authors
The authors have corrected the manuscript according to the reviewer's comments.